# Preweaning Nutrition and Its Effects on the Growth, Immune Competence and Metabolic Characteristics of the Dairy Calf

**DOI:** 10.3390/ani13050829

**Published:** 2023-02-24

**Authors:** Emma M. Ockenden, Victoria M. Russo, Brian J. Leury, Khageswor Giri, William J. Wales

**Affiliations:** 1Agriculture Victoria, Ellinbank, VIC 3821, Australia; 2Faculty of Agriculture, Ecosystem, Food and Forest Sciences, The University of Melbourne, Parkville, VIC 3010, Australia; 3Centre for Agricultural Innovation, The University of Melbourne, Melbourne, VIC 3010, Australia; 4Agriculture Victoria, Bundoora, VIC 3083, Australia

**Keywords:** immune challenge, accelerated milk feeding, heifer calf development

## Abstract

**Simple Summary:**

The rearing of replacement heifers is arguably a dairy farmer’s most important investment and one of the leading expenses on dairy farms. Australian calf rearing recommendations have changed very little in the last 30 years, with recent studies considering them inadequate. Early-life nutrition is widely understood to affect the development of physiological systems in all species; it is therefore essential that effective calf rearing strategies are in place to produce resilient cows to ensure a productive and profitable dairy industry. This experiment followed 20 heifer calves from birth to weaning and investigated the effects of two preweaning nutritional strategies (Low: 4 L or High: 8 L of milk per day) on growth, immune competence, and metabolic characteristics. The physiological systems were compared between treatments in response to an immune challenge in the form of a vaccination. Calves fed a higher milk volume in the preweaning phase had superior growth, immune and metabolic characteristics than calves on the restricted milk diet. Therefore, these results do not support the current industry practice of restricted milk feeding calves. Developments in this area could provide new management approaches improving health, welfare and profitability in the dairy industry.

**Abstract:**

Feeding increased volumes of milk in the preweaning phase has been shown to improve growth, morbidity and mortality rates in calves (*Bos Taurus*). This experiment enlisted 20 Holstein-Friesian dairy replacement calves from birth until weaning (at 10 weeks of age) and assessed the effect of feeding either 4 L (Low) or 8 L (High) of milk per calf per day on their growth, immune competence and metabolic characteristics. The responsiveness of these systems was compared through a vaccination immune challenge. Calves in the High treatment group were significantly heavier from two weeks of age and were 19 kg heavier than calves in the Low treatment group at weaning. Calves in the High treatment group also exhibited greater immune responses, with significantly higher white cell counts and neutrophil counts than calves in the Low treatment group post-vaccination. Calves in the High treatment group also had lower beta-hydroxybutyrate both pre- and post-vaccination, and higher glucose and insulin levels post-vaccination, indicating superior metabolic characteristics. Calves had ad libitum access to lucerne hay (*Medicago sativa*) and a commercial concentrate. Solid feed intakes were mostly the same between treatments, with differences in hay intake only detected at 7 and 8 weeks of age. Results from this experiment are indicative of a positive influence of accelerated preweaning nutrition on growth, immune response and metabolic characteristics.

## 1. Introduction

In Australia, dairy calf morbidity and mortality rates are around twice industry targets (morbidity 23.8% with <10% target; and mortality 5.6% with <3% target) [1]. Recent research has called for a re-evaluation of management strategies for rearing replacement heifers to reduce calf morbidity and mortality rates [1]. Current recommendations prioritise keeping labour and feed costs to a minimum by limiting milk consumption to 10% birth bodyweight daily with the aim of increasing concentrate intake to ultimately promote early weaning [2]. It has been widely demonstrated that calves with access to milk ad libitum consume 50–90% more milk than recommended feeding levels, while those rationed to current recommendations show behaviours associated with hunger [3,4,5]. Early life nutrition is widely understood to affect the development of physiological systems in all species [6]. Optimising early life development in dairy calves has been shown to have both short- and long-term benefits on health, welfare and production. Improved nutrition in the preweaning phase has been shown to increase growth rates, decrease disease susceptibility and mortality, and in turn increase economic benefits with reduced veterinary costs and fewer animal losses [3,5,7,8]. Long-term, these benefits are also evident through increased milk production [9,10,11,12].

The mechanisms behind the health and production benefits of improved preweaning nutrition are not well understood. It has been widely cited that increased nutrition gives rise to greater growth rates in calves, and malnutrition is well known to negatively impact immunity in all species [13]. Yet, as is the case with much calf-related research, the effect of increased milk feeding on pre-weaned calves and their immunity is mixed (either having a positive influence [14] or a negative influence [15] or as reviewed by Verdon [16] and Palczynski et al. [17] for greater detail). It has been previously demonstrated that weaned calves with greater average daily gains (ADG) were categorised as having greater immune responsiveness when faced with an immune challenge [18]. In this case, it would be reasonable to assume that manipulating nutrition in pre-weaned calves would therefore result in a variation in immune function. However, this is yet to be determined. Furthermore, given the suspected differences in metabolic reserves available to calves with varying nutrition, their ability to respond to an immune challenge could be impacted. When evaluating the influence of nutrition on the immune capabilities of an animal, immune challenges, such as those previously mentioned, are considered the gold standard [13], and are generally lacking in calf studies. Further, a considerable number of these experiments are conducted using male calves, when it has been demonstrated that the immune response differs with sex [19].

This experiment examined the effects of two milk feeding strategies (High and Low milk volume) on the growth, immune competence and metabolic characteristics of pre-weaned calves. These physiological parameters were also compared in response to an immune challenge via routine vaccination at 6 weeks of age. The hypotheses tested were: (1) that calves in the High treatment group would have greater body weights at weaning than calves in the Low treatment group; (2) that calves in the High treatment group would have superior immune and metabolic responses to the immune challenge than calves in the Low treatment group.

## 2. Materials and Methods

This experiment was undertaken at the Agriculture Victoria Dairy Research Centre, Ellinbank, Victoria, Australia (38°14′ S, 145°56′ E) from July–October 2020 (Australian Spring). Approval to proceed was granted by the Agricultural Research and Extension Animal Ethics Committee, application number 2020-06.

### 2.1. Experimental Design and Treatments

This experiment used 20 spring-born Holstein-Friesian heifer calves from birth until 10 weeks of age. Calves were all born from heifer dams, within eight days of their expected due date. At birth, one of two treatments was randomly assigned to calves:High: 8 L of milk dailyLow: 4 L of milk daily

Treatments were balanced for calves’ birth weight and estimated Balance Performance Index (BPI) (Australia’s national selection index). Estimated BPIs were calculated prior to birth using the average of the parents’ BPI. If necessary, randomised allocation of treatments was overruled in the final stages of the recruitment process to ensure these parameters remained balanced between treatments.

### 2.2. Animal Management

At birth each calf is fitted with an individual ear tag to allow identification within the herd as part of normal farm practice.

All calves were group-housed within a single pen inside a three-sided shed. The pen dimensions allowed for 4.5 m^2^ floor space for each calf.

Calves were separated from their dam and tube fed 4 L of high quality (>22% Brix) pooled colostrum within 8 h of birth. A second 2 L feed of colostrum was tube fed 12 h after the first colostrum feeding. At 24–48 h of age, passive transfer of immunity was initially screened onsite using serum Brix refractometer methods before enrolment. Passive transfer of immunity was subsequently confirmed through blood serum Immunoglobin G (IgG) concentrations using ELISA methods at a commercial laboratory (Gribbles Veterinary Pathology, Clayton, Victoria, Australia).

Following colostrum tube feeding, the first four milk feeds given to all calves was 2 L of colostrum milk (milk from the 2nd-4th milking post birth), before transition to whole milk from the main vat. Daily milk allocation was divided into two equal feeds and supplied at approximately 0600 and 1630 h. Calves were separated while being fed milk via single-teat feeders to accurately measure each calf’s individual intake. Calves in the High treatment group were gradually accustomed to the higher level of feed to reduce the risk of scours. This was done by increasing the amount of milk offered by 2 L each day (1 L per feed) until they reached the full treatment volume after 2 days. Any refusals were measured, and individual intake was recorded throughout the experiment.

A commercial calf concentrate (Reid Stockfeeds- Gippsland, Trafalgar, Victoria, Australia) and lucerne hay were available ad libitum within the pen, and individual feed intakes recorded via automatic feeders (Gallagher Animal Management Systems, Hamilton, New Zealand). The automatic feeder system identified calves via an electronic ear tag. The system constantly recorded the weight of the feed bin, enabling feed intake to be determined each time a calf visited the feeder. The feed intake of hay and concentrate was reported on a dry matter basis following correction for dry matter concentration.

Calves were weighed using electronic scales at 24–48 h of age to establish birth bodyweight. Bodyweights were then measured weekly for the remainder of the experiment prior to AM milk feeding.

Calves completed the experiment at the commencement of weaning at 10 weeks of age.

### 2.3. Nutritional Measurements

#### 2.3.1. Milk

A representative sample of approximately 20 mL of milk fed to calves was taken twice weekly and sent to a commercial laboratory (HICO laboratories, Korumburra, Victoria, Australia) for milk composition analysis (fat, protein, and lactose concentration). The mean nutritive content of whole milk fed to calves throughout the preweaning period was 3.6 ± 0.58% (mean ± SD) fat, 3.2 ± 0.16% protein, and 4.9 ± 0.09% lactose.

#### 2.3.2. Concentrate & Hay

Representative samples of concentrate and hay were taken each fortnight throughout the duration of the experiment. Half of each sample was dried immediately at 105 °C for 48 h to determine the percentage dry matter (DM). The remaining half was stored at −18 °C until the completion of the experiment. At the time of processing the remaining sample was oven-dried at 60 °C for 72 h before being ground through a 1 mm sieve and sent to a commercial laboratory for wet chemistry analysis (Dairy One Forage Laboratory, Ithaca, NY, USA) [20]. The nutritive characteristics are presented in Table 1.

### 2.4. Immune Challenge

At 42 days of age, each calf commenced an immune challenge using a protocol combined and modified from Aleri et al. [15,17,18]. An interruption of homeostasis occurred via the use of a commercial vaccination for Bovine Respiratory Disease (BRD) (Bovilis MH + IBR; Coopers Animal Health, Macquarie Park, NSW, Australia), and subsequent responses between treatments were compared.

#### 2.4.1. Acquired Immune Response

At day 42, a blood sample was taken from each calf. Immediately following the blood sample, the vaccination was administered according to the manufacturer’s protocol. Additional blood samples were taken at day 50 and day 52 of age to determine the responses. Samples taken on day 42 and 52 involved the collection of 35 mL of blood via jugular venepuncture into four vacutainers (1 × 10 mL lithium heparin, 1 × 10 mL plain, 1 × 10 mL EDTA and 1 × 5 mL fluoride/oxalate tubes). The sample taken on day 50 was a smaller sample and involved the collection of 10 mL of blood into a 1 × 10 mL plain tube. Lithium heparin, EDTA and fluoride/oxalate blood samples were stored on ice prior to centrifugation at 1578 g for 15 min at 4 °C. Prior to centrifugation, a sample of whole blood was extracted from the EDTA tube and sent for immediate haematology analyses (Gribbles Veterinary Pathology, Clayton, Victoria, Australia). Plain tube samples were incubated at 24 °C for 2 h prior to centrifugation at 1258 g for 15 min at 24 °C. Plasma and serum samples were then stored at −20 °C prior to analyses.

#### 2.4.2. Cell Mediated Immune Response

On day 50, hypersensitivity skin fold testing also commenced as per the protocols described in Aleri et al. [18], Aleri et al. [21] and Aleri et al. [22]. On day 50, 0.1 mL of the vaccine was injected intradermally into the right caudal fold of the tail, and 0.1 mL of saline into the left. The caudal skin fold thickness was measured in triplicate prior to injection and repeated 24 and 48 h post-injection using Harpenden skin fold callipers (Creative Health Products Inc., Ann Arbor, MI, USA). The delayed-type hypersensitivity response was corrected using the following formula from Aleri et al. [18], Aleri et al. [21] and Aleri et al. [22]:Increase (mm) = (A − B) − (C − D),
whereby

A = mean test site thickness at 48 h post injection,B = mean test site thickness prior to injection,C = mean control site thickness at 48 h post injection, andD = mean control site thickness prior to injection.

### 2.5. Biomarker Analysis

The first blood sample was taken 24–48 h post birth and was used to confirm passive transfer of immunity, while also providing an experimental baseline of all the biomarkers. The subsequent samples were taken at 42, 50 and 52 days for the immune challenge as previously described. Analyses for the first, second and final samples included the metabolic biomarkers beta-hydroxybutyrate (BHB), non-esterified fatty acid (NEFA) glucose and insulin, and immune biomarkers including a white blood cell (WBC) count and differential and infectious bovine rhinotracheitis (IBR) antibodies. The third blood sample, taken on day 50, measured the IBR antibody titres only.

BHB, NEFA and glucose concentrations were analysed by a commercial laboratory (Regional Lab Services, Benalla, Victoria, Australia) using a Konelab 20XTi autoanalyzer (Thermo Fisher Scientific, Waltham, MA, USA) with assays as follows: BHB assay as per McMurray et al. [23]; NEFA assay kit: Randox Laboratories Ltd., Crumlin, UK; Glucose assay kit: Beckman Coulter Inc., Brea, CA, USA. Insulin assays were developed and conducted at the Assay Centre, Department of Agriculture and Food, Melbourne University. Insulin concentrations were measured in a homologous double antibody radioimmunoassay (RIA). The RIA was performed using purified insulin antiserum raised in guinea pig (Antibodies Australian) and purified bovine insulin for iodination and standard (Sigma–Aldrich, cat#I5500). Briefly, 200 µL or 100 µL assay buffer (0.5% BSA/0.03M sodium phosphate monobasic/0.2M sodium phosphate dibasic/0.1% sodium azide/0.1% triton X) and 200 µL first antibody (1:60,000) diluted in 1:500 normal guinea pig serum (NGPS) was added to duplicate plastic tubes containing either 100 µL standard (1.211–620 µIU/mL) or 200 µL ovine plasma. After preincubation at 4 °C for 72 h, ^125^I iodinated bovine insulin was added (15,000 cpm/100 µL) and the incubation continued for a further 24 h at 4 °C. The antibody-bound hormone was separated from the free hormone by the addition of 200 µL goat-anti-guinea pig serum (GαGPS) (1:200). The tubes were incubated with the second antibody overnight at 4 °C before centrifugation (3500 rpm, 30 min, 4 °C), after which the supernatant was aspirated and the precipitate counted in a gamma counter.

Immune biomarkers were also analysed in commercial laboratories. The white cell count and differential were determined using a Cell-Dyn 3700 autoanalyzer (Abbott Diagnostics, IL, USA) at Gribbles Veterinary Pathology (Clayton, Victoria, Australia). IBR antibodies were analysed at Veterinary Diagnostic Services-Agriculture Victoria (Bundoora, Victoria, Australia) using IBRGBC ELISA kit (IDvet, Grabels, France).

### 2.6. Statistical Analysis

Weekly bodyweight, nutritive intakes and physiological biomarkers were analysed using linear mixed models (LMMs) which were fitted using restricted maximum likelihood, with individual calf as the unit of analysis. In LMMs, the effect of the covariates birth body weight and estimated BPI were fitted as fixed effects, as was the effect of treatments. The effect of calf was fitted as a random effect and used as a residual term, and was assumed to follow a normal distribution with zero mean and constant variance. Histograms of residuals and plots of residuals versus fitted values were examined for normality of distribution with constant variance. Prior to the final analyses, blood insulin measurements were logarithmically transformed to satisfy the assumption of normality with constant variance. The treatment means of blood insulin measurements were back transformed using the bias correction factor $\exp\left(\hat{\mu }+\frac{{\hat{\sigma }}^{2}}{2}\right)$ where $\hat{\mu }$ and ${\hat{\sigma }}^{2}$ is the estimated treatment mean and residual variance, respectively, in the logarithmic scale. All statistical analyses were performed using GenStat 22nd Edition (VSN International, Hemel Hempstead, UK).

Haematology data from one calf in the High treatment group at birth was unable to be used in the analysis due to haemolysis in the sample.

## 3. Results

There was no difference in birth weight between the treatments (Figure 1). From 2 weeks of age onwards, calves in the High treatment group were significantly heavier than calves in the Low treatment group (*p*-value < 0.05). By 10 weeks of age, the calves in the High treatment group were, on average, 19.0 kg heavier than those in the Low treatment group (96.7 kg and 77.7 kg, respectively). From birth, the average daily gain (ADG) for calves in the High treatment group was 0.83 kg/day, compared to 0.56 kg/day for calves in the Low treatment group.

By design, daily milk consumption was higher for calves in the High treatment group than the Low treatment group throughout the experimental period (Figure 2). Apart from the first week, when they consumed 3.9 L, all calves in the Low treatment group consumed the total 4 L of milk offered daily. Calves in the High treatment group consumed almost twice that, drinking an average of 7.9 ± 0.1 L of milk daily from two weeks of age onwards. Prior to this, the High treatment group consumed an average of 5.4 ± 0.1 L and 7.6 ± 0.2 L of milk daily during week 1 and week2, respectively. Over the entire experimental period, the mean amount of milk consumed by the Low treatment group was 279 L, which contained a total ME of 61.5 MJ and 71.6 kg of crude protein (CP). The High treatment group consumed 533 L, containing an ME of 117.3 MJ and 136.5 kg CP throughout the experimental period.

Daily concentrate intake did not differ between the High and Low treatment groups throughout the experiment (Figure 2). Concentrate intakes for the Low treatment group started to increase steadily at 5 weeks of age and the High treatment group a week later at 6 weeks of age. Concentrate intake increased by an average of 0.12 kg DM/calf/day each week from week 5 onwards for calves in the Low treatment group, and 0.10 kg DM/calf/day for the High treatment group. The average intake for both groups remained under 1 kg DM/calf/day throughout the experimental period. The mean amount of concentrate consumed by the Low treatment group was 25.3 kg DM (totaling 3.4 MJ of ME, and 5.5 kg CP) compared to the High treatment group 22.5 kg DM (totaling 2.3 MJ of ME and 4.8 CP) throughout the experimental period.

Hay intakes were significantly higher for the Low treatment group during weeks 7 (*p*-value = 0.034) and 8 (*p*-value = 0.033) only. They were not different between treatments throughout the rest of the preweaning period. Hay intakes were lower than concentrate intakes, with the average daily intake increasing by 0.04 kg DM/calf/day for the Low treatment group and 0.03 kg DM/calf/day each week for the High treatment group. Daily intakes remained below 0.40 kg DM/calf/day throughout the experimental period. Mean hay intake for the Low treatment group was 10.9 kg DM (totaling 0.9 MJ of ME and 1.8 kg of CP) compared to the High treatment group’s 7.8 kg DM (totaling 0.6 MJ of ME and 1.3 kg of CP) throughout the experimental period.

All calves received successful passive transfer of immunity, with IgG levels all above the minimal target level of 10,000 μg/mL between 24–48 h of age.

The WCC and neutrophil count were not significantly different between treatments prior to vaccination (Table 2). However, 10 days post-vaccination, the WCC was significantly higher (*p*-value = 0.048) and the neutrophil count was trending towards significantly higher (*p*-value = 0.054) in the High treatment group. No significant differences were found for monocytes, eosinophils, basophils or the IBR titre between the treatments at birth, pre- or post-vaccination. This is excluding the lymphocytes, where the High treatment group recorded a significantly higher lymphocyte count at birth (*p*-value = 0.050), and the Low treatment group significantly higher pre-vaccination (*p*-value = 0.028).

The High treatment group had significantly lower BHB levels both pre- (*p*-value < 0.001) and 10 days post-vaccintion (*p*-value = 0.012) (Table 2). The High treatment group also had significantly higher glucose (*p*-value < 0.001) and insulin (*p*-value < 0.001) levels 10 days post-vaccination than the Low treatment group, yet not at the sample times prior. There was no difference in NEFA levels detected between treatments at any sample time point.

There was no significant difference in the increase in test site skin fold thickness post-delayed-type hypersensitivity skin fold testing between treatments. Twenty-four hours post-intradermal injections, the Low treatment group measured a corrected increase in skin fold thickness of 5.9 mm, in comparison to the High treatment group who measured a corrected increase of 5.4 mm prior to injections (SED = 0.36; *p*-value = 0.175). At 48 h, the test site skin fold thickness corrected increase was still the same between treatments, with the Low treatment group recording a corrected increase of 6.2 mm versus High treatment with 6.6 mm prior to injections (SED = 0.63; *p*-value = 0.627).

## 4. Discussion

Feeding dairy calves increased volumes of milk in the preweaning phase increased growth rates and improved immune competence and metabolic characteristics without effecting solid feed intake. There are many management factors in calf rearing that contribute to the overall health and development of calves [9], including but not limited to: milk type and volume, forage source and particle size, concentrate texture and ingredients, weaning age and methods. These variations between studies can considerably impact or confound results, and may be responsible for the discrepancies throughout the calf nutritional studies discussed in this paper. Unlike the current experiment, many calf experiments lack immune challenge protocols that are considered the gold standard when evaluating the influence of nutrition on the immune capabilities of an animal [13]. Additionally, many of these experiments are conducted using male calves, and it has been reported in many human and animal studies, including prepubertal calves, that the innate and adaptive immune response differs between males and females [19,24]. The current experiment implemented an immune challenge on female replacement calves as these are the animals that remain within the herd and therefore more accurately reflect the industry. However, given the lack of data for female calves within the literature, studies using male calves have been cited throughout this report; therefore, discretion needs to be used in such instances.

### 4.1. Growth Rates

Feeding increased volumes of milk resulted in heavier calves from two weeks of age until weaning. Calves in the High treatment group demonstrated similar growth paths to other accelerated milk-fed calves [5,8]. However, calves in the Low treatment group were slightly lighter than restricted milk fed calves reported elsewhere [5,8,25]. The smaller average body weight recorded for the Low treatment group exacerbated the difference between treatment bodyweights at weaning. This is thought to be due to the lower concentrate intake from these calves than expected. Given the lack of differences in solid feed intakes recorded in this experiment, all body weight gains in the High treatment group can be attributed directly to increased milk consumption.

### 4.2. Feed Intake

By design, milk intakes for the High treatment group were significantly higher than the Low treatment group throughout the experiment. Calves in the Low treatment group always consumed the 4 L of milk offered daily. Most calves in the High treatment group consumed the increased volume of milk daily. However, two individual calves in the High treatment group would often record refusals bringing the group average below 8 L daily. However, the amount of milk consumed by these High group calves was always more than that of the Low group. Studies offering calves milk ad libitum have recorded daily intakes twice that of current industry practice [3,4] and greater, with intakes of 10 L [7], and even up to 16 L [5] reported from two weeks of age. These ad libitum intakes suggest that even the High treatment group underwent restricted feeding in the current experiment. In which case, the question is posed whether the biological differences observed thus far could be further enhanced with the provision of even greater volumes of milk. The milk intake results for this experiment support the idea that calves can consume far greater amounts of milk daily than the current Australian dairy industry standard [2].

Concentrate intakes were not significantly different and were considered low for both treatment groups. Typically, restricted milk-fed calves are shown to have higher solid feed intake as compensation for being on limited milk [3,7,8]. It is this common principle that forms the basis of the current calf feeding recommendations in Australia. However, given the differences in energy availability between milk and concentrates, calves cannot consume enough concentrate to match the energy source from milk, and the recommendations are now being revised [7,26]. Both treatment groups recorded low solid feed intakes; while the reduced solid feed intake was expected in the High treatment group, the reason for the reduced solid feed intake in the Low group is unclear. Due to the lack of compensatory concentrate intake by the Low treatment group, all differences in ADG, immune and metabolic biomarkers found in this experiment can be attributed to the greater overall increased energy intake from the higher volume of milk fed to calves in the High treatment group.

Hay intake increased from two weeks of age yet remained the lowest contributor to overall daily feed intake. Intakes of hay in this study were slightly higher for both treatment groups than previously reported [7,27], this could be due to the type of hay offered. The current study provided lucerne hay as the forage source, which has been noted as highly palatable leading to increased intakes than other forage sources [27,28]. The suitable provision of hay and its effect on the development of young calves is contrary [29]. Like milk, previous recommendations have advised limiting forage supplementation to avoid non-nutritive gut fill and promote more energy-rich concentrate intake [2]. On the other hand, forage intake early in the preweaning phase is considered vital for both physical and chemical rumen development [29,30]. While the current intakes of hay were only slightly higher than those reported by others [7,27], it would not be unreasonable to question its contribution to the current experiment’s low concentrate intakes. While hay intakes were significantly higher for the Low treatment group during weeks 7 and 8 alone (week 7: *p*-value = 0.034; week 8: *p*-value = 0.033), it is unlikely that these two weeks of increased intakes had much influence on overall results. Furthermore, like concentrate intakes, the differences in ADG, immune and metabolic biomarkers detected in this experiment were not influenced by hay intake alone, but most likely by the overall increase in energy provided to calves in the High treatment group.

### 4.3. Immune Biomarkers

Immune biomarkers analysed were not significantly different between treatments at birth or prior to vaccination. However, there was a small difference in lymphocyte count at birth and pre-vaccination between the High and Low treatments. While significantly different, all results remain within the normal biological range [31], so the biological significance of this result is minimal.

Post vaccination, the WCC and neutrophil count were both significantly higher in the High treatment group when compared to the Low treatment group (*p*-value = 0.048 and *p*-value = 0.054 respectively). This result indicates that in this experiment calves in the High treatment group have an improved immune response due to the increased level of nutrition. In contrast, Foote et al. [32] found no difference in the WCC differential of calves at different preweaning growth rates after a vaccination. Furthermore, Obeidat et al. [33], reported higher neutrophil activity in calves fed a lower plane of nutrition. While these two experiments are some of the few to include a form of immune challenge, they differed from the current experiment as they were both conducted using male calves. However, like both of these studies we conclude that further research is required to understand the mechanisms behind our findings, and what implication they have on the overall immune function of calves.

The IBR (vaccine specific antibody) titre was not different between the two treatments. Previous experiments undertaking vaccine immune challenges have found relationships between vaccine titres and greater ADG using similar challenge protocols with a different vaccine in older calves [18]. The IBR vaccine was selected for use in this experiment because it was a novel vaccine that was not routinely used as part of the farm’s vaccine protocol. However, at birth, positive IBR titres were evident in the blood of some calves, suggesting pre-exposure to IBR. Further, post-vaccination, one calf returned a negative titre and two others doubtful S/N% titres. Given the positive immune responses found in the WCC and neutrophil count post-vaccination we are confident an adequate immune response was generated from the vaccine. The lack of response from some of the IBR titre results potentially indicate the need for further consideration of protocols including vaccine type and duration between blood tests in future research.

In more general terms, the effect of increased nutrition on the immune status of calves is mixed. Supportive of the previously discussed immune biomarker results, calves on an increased plane of nutrition have reported improved immune status with fewer incidences or faster resolution of diarrhea [8,14]. On the other hand, Borderas et al. [5] and Gerbert et al. [34] reported no difference in health or immune status. Quigley et al. [35] and Curtis et al. [15] reported that calves fed greater volumes of milk recorded more incidences of diarrhea requiring veterinary treatment. While our increased WCC and neutrophil count results indicate superior immune competence in calves fed high volumes of milk, the lack of difference from the IBR titre results are potentially contrary. Further research with greater sample size and vaccine protocol consideration would assist in confirmation of the effect of feeding a greater volume of milk on the immune status of calves.

### 4.4. Metabolic Biomarkers

Metabolic biomarkers did not differ between treatments at birth. However, at 42 days of age, prior to the immune challenge, calves in the Low treatment group already had significantly higher levels of BHB (*p*-value =< 0.001), which was maintained post vaccination when both groups reported increased levels of BHB (*p*-value = 0.012). NEFA levels, however, did not differ at any point between the treatments. Increased circulating levels of NEFA and BHB are a result of the increased mobilisation of body fat reserves in response to a negative energy balance. In mature cows this can be an indication of metabolic disease [36]. In well-fed pre-weaned calves, the mobilization of body reserves is not likely, and so elevated levels of BHB and NEFA are not typically reported in the literature. While the BHB levels did differ between treatments, all results in the experiment were within normal diagnostic limits [37]. The level of nutrition is commonly reported as having no influence on the level of NEFA in pre-weaned calves [14,32,38]. However, attempts have been made to use higher levels of BHB as a proxy indicator for increased concentrate intake and rumen development with mixed results. Khan et al. (2011) and Byrne et al. (2017) both found it a suitable indicator, whereas Suarez-Mena et al. (2017) did not. Calves’ BHB levels are also understood to increase with age [30,38,39]. BHB levels in the Low treatment group did increase with age and concentrate intake, supportive of previous trends in the literature. However, the results for the High treatment group seem to more closely resemble metabolic trends in the adult cow, and therefore may query the appropriateness of the current theories of calf BHB trends. BHB levels were lowest pre-vaccination, when obviously age and concentrate intakes were higher than at birth. Further, BHB levels were highest at times of increased metabolic stress imposed by the immune challenge and birth, and lowest at a time when metabolic stress on the animal would be considered lowest for the experiment. Additionally, age and concentrate intakes were not different between treatments, and therefore it is unlikely that these factors influenced the differences detected, indicating that the differences in BHB levels observed between treatments were potentially a direct result of the level of nutrition provided by milk.

Glucose and insulin levels were significantly lower for the Low treatment group post-vaccination (*p*-value =< 0.001 and *p*-value =< 0.001 respectively). Pre-vaccination, Low treatment group calves also reported a trend towards lower glucose and insulin levels, although not a significant one (*p*-value = 0.056 and *p*-value = 0.063 respectfully). Similar nutritional effects were observed by Byrne et al. [38], where glucose levels were higher in calves fed greater volumes of milk. Glucose levels remained constant for the High treatment group both pre- and post-vaccination, whereas glucose levels continually reduced throughout the experiment for the Low treatment group. In calves, glucose levels are understood to decrease with age, as they transition from a monogastric animal into a ruminant and energy sources shift [39,40]. Like BHB, glucose levels in the restricted fed calves support this theory, but those in the High treatment group do not. Glucose levels were not affected by the immune challenge in the High treatment group like the Low group. Considering the differences in immune responses previously discussed, this could indicate greater metabolic pressure for an inferior response from the Low milk-fed calves, as well as fewer energy reserves available for maintenance and growth. Unlike glucose, the difference in insulin levels between treatments was not solely due to the reduction in insulin levels from the Low treatment group, but mostly due to a simultaneous increase in insulin levels from the High treatment group. Insulin levels increased post-vaccination for the High treatment group, with little change (slight reduction) in circulating glucose levels. While its exact role is not completely understood, insulin is known to play a modulatory role in immunity and the inflammatory response [41,42]. Higher levels of insulin have been associated with lower disease severity in human patients suffering various inflammatory diseases [41]. Despite insulin initially being examined as a metabolic biomarker, these results may have further contributed to the evidence supporting the superior immune response generated by calves fed a high volume of milk.

From these results it would not be unreasonable to question the existing theories of biomarkers as indicators of calf development. Calves’ energy demands for maintenance and growth rise with age and increasing size. As the energy supply from milk is restricted and compensation of energy from solid feed intake is increasingly insufficient, it may be plausible that they become more metabolically strained with age. More research is needed to re-evaluate the various biomarkers and the effect increasing preweaning nutrition has on their trends.

### 4.5. Delayed-Type Hypersensitivity Skin Fold Testing

Reactions to the delayed-type hypersensitivity skin fold tests were not different between treatments. Like the current experiment, Aleri et al. [18] and Aleri et al. [21] also did not find significant correlations between ADG and hypersensitivity skin fold testing responses in older calves or cows. Foote et al. [32] did find differences in skin welt size in calves at various growth rates; however, it did not correlate with ADG. While nutrition seems to affect some areas of immunity, these results further support that cell-mediated immunity is not influenced by nutrition.

## 5. Conclusions

The findings from this experiment stipulate that feeding calves increased volumes of milk in the preweaning phase has a positive influence on growth, immune competence and metabolic characteristics. An improved immune response to the vaccination was observed in the High treatment, where there were higher WCC, neutrophil count and insulin levels compared to the Low treatment. The greater metabolic reserves available for these calves allows greater immune responses with additional energy available for maintenance and growth. The results from this experiment do not support the common industry practice of restricted milk feeding of calves.

How to best balance the supply of energy from liquid and solid feed is yet to be determined, alongside an understanding of their interactions on developing systems to improve calf nutrition. Future work is needed to confirm these uncertainties in addition to the preservation of such traits (i.e., growth, immunity and metabolism), and the ongoing effect this has on future milk production. Furthermore, information on the contribution of the reported improved immune response on reducing morbidity and mortality rates in dairy calves would benefit the industry and recommendations moving forward.

## Figures and Tables

**Figure 1 animals-13-00829-f001:**
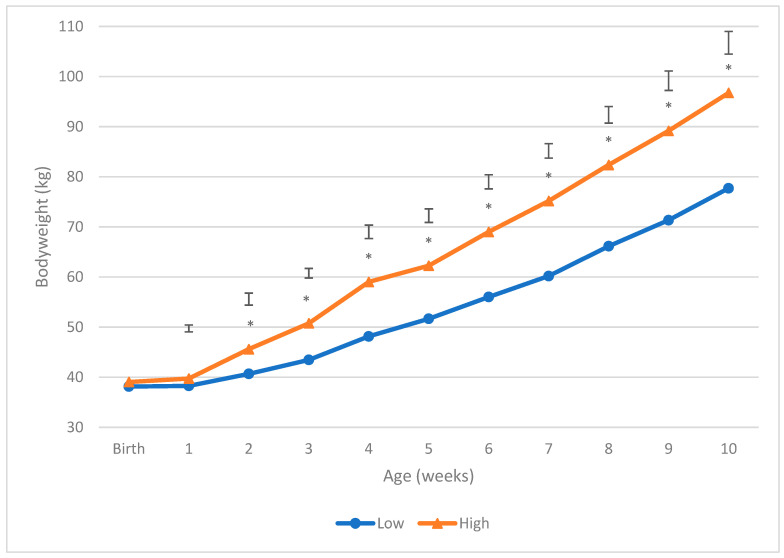
Body weight (kg) of calves offered 4 L (Low) vs. 8 L (High) of milk daily. Asterisks * indicate significant difference (*p*-value < 0.05). The least significant difference (LSD) is given with vertical bars at each week.

**Figure 2 animals-13-00829-f002:**
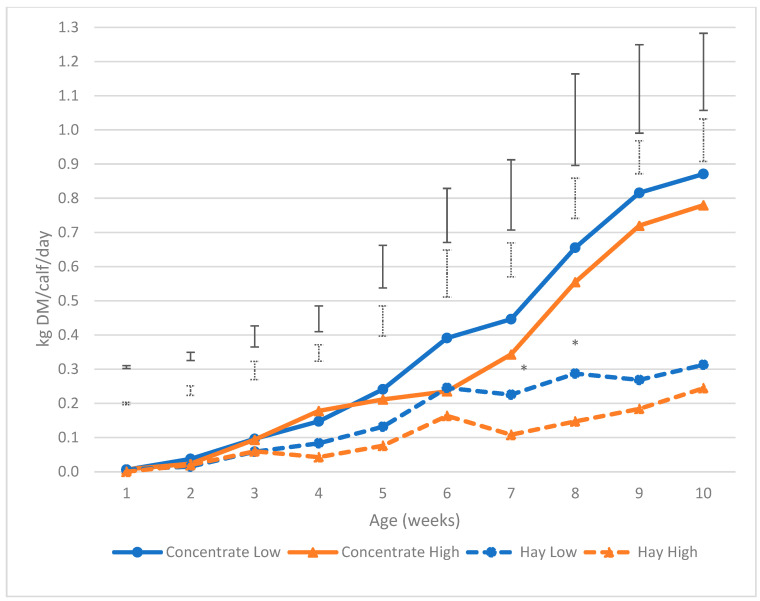
Average daily concentrate (kg DMI/calf/day) and hay (kg DM/calf/day) consumption of calves offered 4 L (Low) vs. 8 L (High) of milk daily. Asterisks * indicate significant difference (*p*-value < 0.05). The least significant difference (LSD) is given with vertical bars at each week (concentrate LSD lines appear above hay).

**Table 1 animals-13-00829-t001:** Mean nutritive characteristics of feed offered during the preweaning phase ^1^ (Crude protein (CP), acid detergent fibre (ADF), neutral detergent fibre (NDF), lignin, non-fibre carbohydrates (NFC), starch, crude fat (CF) and metabolizable energy (ME)).

	CP	ADF	NDF	Lignin	NFC	Starch	CF	Ash	ME ^2^
Concentrate	21.7	8.0	14.1	3.2	52.5	33.5	3.7	8.1	13.2
Lucerne hay	16.5	38.7	47.9	9.8	22.9	0.4	1.8	11.0	8.1

^1^ All values are % DM unless otherwise stated; ^2^ MJ/kg DM.

**Table 2 animals-13-00829-t002:** Immune and metabolic biomarkers of calves offered 4 L (Low) vs. 8 L (High) of milk daily at birth, pre-vaccination, 8 days post-vaccination and 10 days post-vaccination (WCC, white cell count; IBR, infectious bovine rhinotracheitis; BHB, beta-hydroxybutyrate; NEFA, non-esterified fatty acid).

Time	Biomarker	Low	High	SED	*p*-Value
**Birth**					
	WCC ^1^	9.4	12.4	2.14	0.170
	Neutrophils ^1^	6.9	9.1	1.85	0.246
	Lymphocytes ^1^	1.9	2.6	0.33	0.050 *
	Monocytes ^1^	0.3	0.5	0.12	0.108
	Eosinophils ^1^	0.1	0.1	0.06	0.443
	Basophils ^1^	0	0	-	-
	BHB ^2^	0.04	0.04	0.013	0.733
	NEFA ^2^	0.30	0.27	0.059	0.586
	Glucose ^2^	8.12	8.02	0.493	0.852
	Insulin (µIU/mL)	15.72	22.61	-	
	loge Insulin	2.66	3.02	0.198	0.085
**Pre-vaccination**					
	WCC ^1^	7.3	7.7	0.75	0.574
	Neutrophils ^1^	2.6	3.8	0.62	0.081
	Lymphocytes ^1^	4.3	3.6	0.30	0.028 *
	Monocytes ^1^	0.4	0.4	0.09	0.728
	Eosinophils ^1^	0.2	0.0	0.09	0.088
	Basophils ^1^	0	0	-	-
	IBR Titer (S/N%)	17.0	12.6	8.14	0.593
	BHB ^2^	0.07	0.02	0.010	<0.001 *
	NEFA ^2^	0.16	0.15	0.027	0.740
	Glucose ^2^	6.01	6.80	0.383	0.056
	Insulin (µIU/mL)	12.23	32.62	-	
	loge Insulin	1.92	2.90	0.490	0.063
**8 days post-vaccination**					
	IBR Titer (S/N%)	18.5	14.5	8.45	0.642
**10 days post-vaccination**					
	WCC ^1^	9.3	11.9	1.21	0.048 *
	Neutrophils ^1^	5.2	7.6	1.18	0.054
	Lymphocytes ^1^	3.6	3.5	0.49	0.783
	Monocytes ^1^	0.4	0.6	0.16	0.252
	Eosinophils ^1^	0.1	0.1	0.04	0.460
	Basophils ^1^	0	0	-	-
	IBR Titer (S/N%)	18.6	16.2	8.59	0.785
	BHB ^2^	0.11	0.06	0.018	0.012 *
	NEFA ^2^	0.20	0.15	0.032	0.095
	Glucose ^2^	5.46	6.72	0.183	<0.001 *
	Insulin (µIU/mL)	11.08	40.18	-	
	loge Insulin	2.25	3.54	0.252	<0.001 *

^1^ Values are ×10^9^/L, ^2^ Values are in mmol/L, Asterisks * indicate significant difference (*p*-value < 0.05).

## Data Availability

Not applicable.

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
