# Peer review of "Preweaning Nutrition and Its Effects on the Growth, Immune Competence and Metabolic Characteristics of the Dairy Calf"

_animals, 2023, doi:10.3390/ani13050829_

Round 1
Reviewer 1 Report
General
This is a simple but interesting study that addresses the issue of calf rearing strategies that is relevant to current on farm practices. The relative novelty of this work is related to the measurements of immune response to the different rearing strategies. There are only a few minor comments for recommended changes to the paper in the sections below.
Introduction
Line 76 – missing ‘and’ in this sentence
Materials and Methods
Line 91 – Can you explain the randomisation process as you indicate in line 94 that the treatments were ‘balanced’ for birthweight and BPI -hence not completely randomised
Line 150 and line 170 – does it require all three references to determine the method used. One reference or a summary of combined aspects of a method would be useful.
Results
Line 246 – Figure 2. This figure is not really needed. It doesn’t provide anything that the text doesn’t describe adequately.
Lines 254 to 266 – while what you have presented is statistically correct - there appear to be a numerical impact of treatment on DM intake, with small experimental numbers possibly making it difficult to identify statistical significance for this response.
Discussion
Line 303 – ‘plagues’ is an emotive word – perhaps something like ‘is commonly found in’ – and perhaps include a few references.
Line 330 – ‘of’ missing in this sentence
Line 335 – Shouldn’t start a sentence with ‘And’
Line 342 – don’t need ‘founding’ and ‘basis’ perhaps get rid of ‘founding’
Conclusion
Line 482 - -used ‘animals’ and ‘calves’ - Best to be consistent
Line 488 - -what do you mean by ‘such traits’ - Please reword this sentence
Reviewer 2 Report
Comments
1; there appears to be an editing error in the abstract -Beta hydroxy butyate is lower in the high groups.
2; I suggest the abstract should note that weaning age is 10 weeks
3; I wonder if the total amount of hay and concentrates over the 10 weeks should be mentioned? The authors rightly concentrate on the daily intakes.
Reviewer 3 Report
In the manuscript ID animals-2228481, the authors investigated the effects of milk feeding levels on dairy heifer calves in Australia on feeding performance and immunity. This experimental data is useful information for the Australian and global dairy cattle industry. However, this manuscript needs some improvements.
1) Authors did not clearly indicate the scientific novelty of the study in the introduction. What's new about this study is the use of females to test immunity?
2) Authors should indicate the range of P-values that determine significant differences in statistical analyses.
3) Authors should present the energy and protein intake of heifer calves in Results and Figures or Tables.
L280 Is this description correct? In Table 2, this difference between treatments does not seem to disappear pre-vaccination.
L291-299 Where are the results of the delayed hypersensitivity skin fold testing shown?
L302-303 Is this description correct? Concentrate intake had no effect, but an effect on hay intake has been shown in L261-262 and Figure 3.
L385-408 Due to your hypothesis that the immune response differs with sex (L309-313), the references should be considered separately for male and female experiments.
Reviewer 4 Report
Dear Authors,
Thank you for submitting this paper that explores the effect of milk provision on the effects of calf growth and immune status. This is a practical, applied study that helps to address some important questions in Australian dairy cattle farming. As such, there is some merit to the paper.
At current however, there seem to be some revisions required in the manuscript to ensure the work is scientifically robust. I have attached the PDF version of the manuscript with specific comments. Additionally, please consider the following points:
1. Make sure the statistical test outputs are reported when you are stating significance.
2. Discussion. Currently the discussion is quite confused, especially when you discuss previous studies. Try to unpick the impact / effects of different studies on cattle growth / immunity. Why are the results confounded? Is it related to the amount of milk provided, for example?
